# Environmentally Specific Servant Leadership and Employee Workplace Green Behavior: Moderated Mediation Model of Green Role Modeling and Employees' Perceived CSR

**Fan Gu * and Jiaqi Liu**

Business School, China University of Political Science and Law, Beijing 100088, China
* Correspondence: gufan@cupl.edu.cn

**Abstract:** While employee workplace green behaviors (EWGBs), which contribute toward goals of organizational sustainability, have developed as an important research topic in the last decade, environmentally specific servant leadership (ESSL) is considered an important enhancer of employees' green behaviors. From a social learning perspective, we developed a theoretical model to explore the mediating role of green role modeling and the moderating effect of employees' perceived CSR on the relationship between ESSL and EWGB. In order to test the hypotheses, we adopted a two-wave research design and collected survey data from 512 employees from eight companies in Shandong Province, China. Structural equation modeling and the PROCESS macro for SPSS were utilized to analyze the data from the survey. The results of the quantitative analysis suggest that ESSL has positive impact on both employees' in-role green behavior and extra-role green behavior. Meanwhile, green role modeling has a mediating effect on the relationship between ESSL and EWGB. Moreover, employees' perceived CSR moderates the mediating relationship between ESSL and EWGB through green role modeling. The findings indicate that organizations should promote managers' environmentally specific servant leadership and establish their own CSR policies and practices in order to motivate employee workplace in-role and extra-role green behaviors, which, in turn, contribute to sustainability, environmental protection and societal development. Overall, the theoretical and practical implications of our findings for the research on ESSL and EWGB are also discussed.

**Keywords:** employee' workplace green behaviors; environmentally specific servant leadership; green role modeling; employees' perceived CSR; sustainability

## 1. Introduction

There is no doubt that environmental issues are not only related to our health, but also to financial development and urbanization. However, some scholars have pointed out that the existing standards of environmental regulations are not sufficient to reduce environmental degradation [1–3]. The current trend of natural environment degradation is becoming increasingly more obvious, which is of great concern [4,5]. Because of the demand for progressively stricter national environmental policies (e.g., public disclosure of enterprise environmental performance) and organizational stakeholders (e.g., employees, consumers and regulators), organizations across the world, in pursuit of sustainable development, are beginning to implement environmental policies and practices in their organizations [6,7]. The implementation of corporate green policies require the cooperation of employees, and employees' workplace green behaviors (EWGBs), to a certain extent, affect corporate environmental initiatives [8,9]. However, the inertia of most organizational employees toward environmental issues has been hindering the advancement of corporate environmental strategies [10]. In this context, exploration of the factors that influence employees' pro-environmental behavior is of increasing interest to organizations

and researchers [9,11,12]. Organizations have begun to implement socially responsible policies and practices [6], which require their employees to behave in "green" ways in the workplace [8,13]. A number of existing studies have documented that EWGBs not only contribute to organizational sustainability and the conservation of natural resources [4,6,10], but also improve the environmental and financial performance of an organization [14–16]. Thus, both scholars and practitioners have paid great attention to the factors that could give rise to these pro-social behaviors, among which environmentally specific servant leadership (ESSL) is considered to be an important enhancer [10,17].

Environmentally specific servant leadership is defined as "providing direction for, empowering and developing people to be pro-environmental citizens, and demonstrating humility, authenticity, interpersonal acceptance and stewardship towards employees' pro-environmental contributions" [12]. Robertson and Barling [13] were the first to extend the concept of servant leadership, which focuses on facilitating followers' well-being, development and success (e.g., [18,19]), to the environmental domain; then, an increasing number of studies began to examine the characteristics of ESSL and its impact on various employees' outcomes. Environmentally specific servant leaders, who are regarded as a role model with green values, serve and help others, e.g., their subordinates, become devoted to achieving the green goals of their organizations and society [17,20]. Despite existing research on ESS leadership, an understanding of the mechanism explaining the relationship between ESS leadership and employees' relevant behaviors, such as workplace green behaviors, remains lacking.

To fill this knowledge gap, this paper intends to empirically examine the relationship between ESS leadership and employee workplace green behaviors by utilizing a social learning perspective. According to social learning theory, individuals change their behaviors by imitating and learning the behaviors of role models [21]. Employees' perception of their leaders as role model plays an important role in their social learning process, and the existing research demonstrates that this perception can significantly influence their own ethical behaviors (e.g., [22]). Some studies use social learning theory to explain why employees learn green behaviors from their leaders [23], but these studies have not yet identified the mechanism that influences employees' social learning from an ESS leader. ESS leaders strive to promote and foster green behaviors among employees through a series of green conducts, such as green empowerment, green value creation, providing assistance to followers regarding achieving environmental goals, putting the environment first, and ethical behavior toward the environment [17], which are green role models for their subordinates, and subsequently strive to increase their EWGB. Moreover, employees' perceived CSR moderates this relationship because employees would pick up on these signals and learn from those CSR-related organizational actions and policies and then engage in more EWGB.

Overall, by constructing a conceptual model and conducting an empirical study, we sought to contribute to the literature in the following three ways. First, this study identifies and examines employees' perception of a green role model as a social learning mechanism linking ESSL to EWGB. By exploring the mediating effect of an employee's perception of green role models on this relationship, this study deepens the understanding of the mechanisms that shape EWGB by adopting a social learning perspective. Second, based on scholars' categorization of EWGB into in-role and extra-role green behaviors in previous studies, this paper explores the impact of ESSL on not only in-role but also extra-role green behaviors. Employees have varying degrees of discretion in their in-role and extra-role behaviors at the workplace [24]; thus, the same factors may have distinct impacts on employee in-role and extra-role green behaviors [25]. Our study aims to investigate whether employees' imitation of green role models affects employee in-role and extra-role green behaviors differently. Third, this study enriches the literature on ESSL and EWGB by clarifying when ESSL can be more effective at promoting EWGB. By integrating the mediating effects of an employee's perceptions of green role models and the moderating effects of employees' perceived CSR, this study provides a greater elucidation of how and

under what conditions ESSL is more effective at promoting EWGB. Thus, our conceptual model is presented in Figure 1.

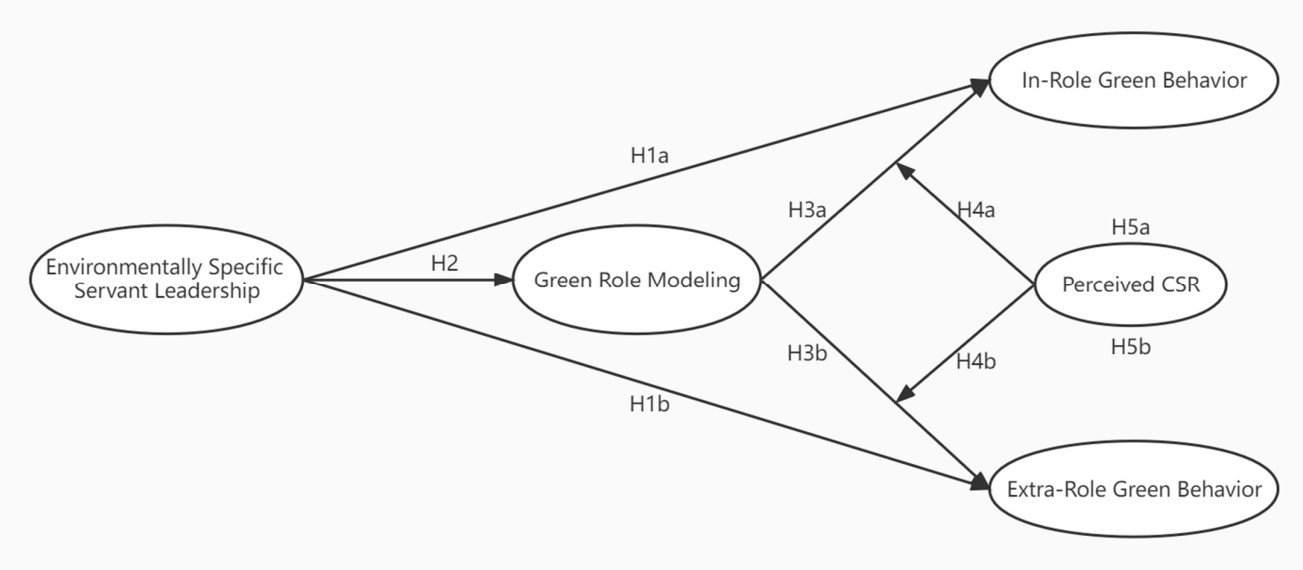

**Figure 1.** Proposed theoretical model.

Relying on the theoretical foundation provided by Bandura's social learning theory, our primary research goal was to construct an integrated model by utilizing ESSL and CSR to explore the formation of employees' workplace green behaviors. The specific objectives were to: (1) examine whether employees' green behavior learning processes are consistent with the role model learning perspective proposed in Bandura's social learning theory; (2) examine whether ESSL affects the extent to which employees view their leaders as green role models; (3) explore the moderating effect of CSR in the relationship between green role modeling and employee workplace in/extra-role green behavior. Different from previous studies, our study takes the four-step process of employee social learning as an entry point to investigate the mechanism of employees imitating the green behavior of their leaders. As a result, our study can provide a new research direction to examine EWGB and provide strategic advice to organizations on human resource management to foster green behaviors among employees and to better achieve their own sustainable development goals.

## 2. Theoretical Background and Hypothesis Development
### 2.1. Environmentally Specific Servant Leadership and Employee Workplace Green Behavior

Employee workplace green behavior (EWGB) is defined as "actions and behaviors that employees engage in that are linked with and contribute to or detract from environmental sustainability" [4]. Previous studies categorized employee workplace green behaviors into two types (e.g., [25,26]): in-role green behaviors, which refer to green behavior that is required or formally rewarded by the organization for a specific job [26], and extra-role green behaviors, which refer to employees' discretionary green behaviors that are neither required nor formally rewarded [11]. How green behaviors are categorized into in-role behaviors and extra-role behaviors is mainly dependent on the type of organization and what the organization expects from its employees [11,25]. Previous studies have given much more attention to employees' extra-role green behaviors, and focused less on in-role green behaviors [9,10]. Therefore, fully studying both in-role and extra-role green behaviors is necessary to gain a more comprehensive understanding of employee workplace green behaviors [25].

The existing literature has documented the positive impact of ESSL on employee green behaviors by drawing on the conservation of resources theory (with climate for green creativity as a mediator [27]), self-determination theory (with psychological empowerment

and autonomous motivation for the environment as a sequential mediator [10]), and social identification theory (with employee environmental engagement as a mediator [20]). Recent studies have demonstrated that employees learn from their supervisors and adopted social learning theory to explain the effect of ESSL on employees' attitudes and behaviors (e.g., [17]). Worth noting is that some researchers believe that the social learning process is important in understanding the impact of leadership styles on employee green behaviors [5]. Following this thought, this research aims to adopt a social learning perspective to examine and clarify the impact of ESSL on EWGB.

Behaviors resulting from corporate policies requiring employees to behave in "green" ways at the workplace could be regarded as in-role green behavior. For instance, some companies require their employees to ensure that wastewater and exhaust gas generated during the production process do not pollute the environment, which is consistent with an organizational green-oriented regulation. In a working setting, ESS leaders provide knowledge and skills relating to environmental activities and sufficient green guidance to their subordinates [12], to help them develop eco-initiatives and to foster pride in their pro-environmental contributions [20]. Thus, in terms of social learning theory, with the support from their supervisors, employees learn how to better behave in a green way, resulting in better in-role green behaviors.

On the other hand, employees' extra-role green behaviors, which are cryptic in nature, refer to behaviors that are beyond the confines of the job description, such as turning off computers and lights that are not in use [25]. The majority of employees' green behaviors within organizations should be classified as extra-role green behaviors [26]. ESSL could foster proactive employee green attitudes [12,28] and create pro-environmental values among subordinates [29]. Hence, by learning from their supervisors, employees can form green attitudes and values and are more likely to engage in green behaviors that go beyond their job requirements, i.e., extra-role green behaviors, and beyond in-role green behaviors [25]. Thus, we predict the following:

**Hypothesis 1 (H1):** *Environmentally specific servant leadership is positively related to (H1a) employee workplace in-role green behavior and (H1b) employee workplace extra-role green behavior.*

*2.2. Environmentally Specific Servant Leadership and Employee's Perception of Green Role Modeling*

A role model is defined as "a cognitive construction based on the attributes of people in social roles an individual perceives to be similar to him or herself to some extent and desires to increase perceived similarity by emulating those attributes" [30]. However, simply being exposed to potential models does not guarantee that a person will take them as a role model and subsequently learn from them [21,22]. Therefore, we address the observational learning process to describe how employees view their environmentally specific servant leaders as green role models.

From the perspective of social learning theory, individuals select a role model and learn attitudes and behaviors from the role model through four sub-processes [31]. The first step is the attention process, during which an effective role model learning process can occur if the individual is able to keep their attention on the role model themselves as well as their behaviors [32,33]. Weiss [34] claimed that the personal and behavioral characteristics of the employees' leader can have a significant impact on the learning process. Servant leaders who are humble, approachable, trustworthy and dependable are believed to attract the attention of employees [32,35] and to enhance the connection between employees and leaders. Thus, individuals will be more likely to be impressed and attentive to the behavior of their ESS leaders.

Nonetheless, the individual cannot be influenced to a large extent if the individual does not remember the behaviors of the role model and translate them into action. Thus, according to Bandura [31], following the attention process, individuals encode their role model's traits, characteristics and actions into symbolic forms (e.g., memories or verbal coding) in the retention process and further transform these memories or verbal coding into

a certain behavior in the reproduction process. Environmentally specific servant leaders who act consistently and predictably according to organizational green policies and national environmental regulations deliver meaningful messages more easily to their subordinates. Therefore, subordinates could easily encode the behaviors of and retain memories of their ESS leaders. As such, these symbols and codes regarding green standards are impressed upon employees and trigger their green behaviors.

In the fourth step, which is named the motivation process, an individual's motivation, which drives their actions, is reinforced in the observational learning process [31]. ESS leaders are inclined to offer useful resources, such as knowledge and value, to help subordinates contribute to the sustainability of the organization and society [12]. Furthermore, green empowerment from an ESS leader could increase an employee's sense of self-efficacy and trust. Correspondingly, employees are more likely to be motivated to behave in "green" ways with the support and encouragement of their supervisors. Overall, in terms of the four sub-steps of the social learning process, ESS leaders become good green role models for employees to emulate. Thus, we propose the following:

**Hypothesis 2 (H2):** *Environmentally specific servant leadership is positively related to green role modeling.*

### 2.3. The Mediating Role of Leader's Green Role Modeling in the Relationship between ESSL and EWGB

As outlined above, by describing the four sub-processes of employees' social learning processes from ESS leaders, we elaborate that ESS leadership increases the extent to which an employee regards their ESS leader as a green role model. According to social learning theory, employees are able to obtain better understanding of what green behaviors are expected from their green role models and will be more inclined toward carrying out behaviors that are similar to those exhibited by their green role models [21,22,36]. As Luu [20] indicates, ESS leaders provide sufficient green knowledge and skills to their employees and even tend to transform their employees into other ESS leaders. Once an employee notices their supervisors' servant behaviors, an employee can emulate their supervisors to accomplish their in-role green tasks properly and to exhibit more green behaviors that are beyond their job requirements. Moreover, ESS leaders tend to improve their green performances at both the individual and team levels first by shaping the green climate in their work team and then by cultivating green values in their employees [12]. Given that employees' green behaviors are one of the most fundamental ways of contributing to organizational green performance, their in-role and extra-role green behaviors should be the outcome of employees' emulation of their green role models. In this way, employees' imitation of their ESS leaders might lead to increased green behaviors through the learning process. Thus, we predict the following:

**Hypothesis 3 (H3):** *Green role modeling mediates the relationship between environmentally specific servant leadership and employee workplace in-role green behavior (H3a), employee workplace extra-role green behavior (H3b).*

### 2.4. The Moderating Role of CSR Perception in the Relationship between Leader's Green Role Modeling and EWGB

CSR, which is defined as "context-specific organizational actions and policies that take into account stakeholders' expectations and the triple bottom line of economic, social and environmental performance" [37], is crucial to organizational sustainability [38]. Organizations tend to make contributions to environmental protection and society development through their CSR-related solid actions [39]. Recently, the focus of CSR studies had been shifted from the impact of CSR on organizational performance [40–42] to the exploration of how perceived CSR influences employees' attitudes and behaviors [43,44]. Existing studies have demonstrated a positive relationship between perceived CSR and various employee outcomes, such as job satisfaction, organizational commitment and

identification, organization citizenship behavior, employee job performance and employee creativity [45–51].

In this study, we suggest that employees' perceived CSR moderates the relationship between green role modeling and EWGB. More specifically, we think that the relationship between green role modeling and EWGB is stronger for those who perceive a high level of CSR in their organizations as opposed to employees who perceive a low level of CSR. Employees' CSR perceptions affect this relationship for the following reasons. First, in terms of a social learning perspective, organizations that engage in CSR acts could be good green role models from which employees can learn green values, knowledge, and skills. Employees are inclined to internalize the values and beliefs of their organization as their own and to engage in actions consistent with these values and beliefs [52,53]. Organizations that adopt CSR policies reportedly strive to protect the environment and care about societal benefits [54]; consequently, employees are encouraged to protect the environment by exerting green behaviors [9,55]. Second, the implementation of CSR leads to a caring and supportive organizational climate [50], which predominately triggers ethical behaviors, such as EWGB [56].

According to social learning theory, we propose that ESSL is more related to EWGB when employees perceive a high level of organizational CSR. In the context of a high level of CSR perceptions, employees who learn green values and behaviors from their ESS leaders could learn more green skills from CSR policies and can obtain green resources from the organization as well, which could help them exhibit better green behaviors. In contrast, employees who perceive a low level of CSR have fewer opportunities to learn from their organization, which perhaps leads to ordinary green behaviors. Thus, we propose the following:

**Hypothesis 4 (H4):** *Employees' perception of CSR moderates the relation between green role modeling and employee workplace in-role green behavior (H4a), employee workplace extra-role green behavior (H4b), such that the relationship is stronger for employees with a high perception of CSR than for employees with a low perception.*

Based on Hypotheses 3 and 4 above, employees' perception of CSR may affect the strength of the indirect relationship between ESSL and EWGB. In other words, we expect that the indirect effect of ESSL on EWGB through green role modeling differs between high and low perceptions of CSR. Specifically, we suggest that ESSL will be more likely to be seen as a green role model by employees, and employees will be more inclined toward demonstrating in-role and extra-role green behaviors at the workplace when they experience a high level of perceived CSR. That is to say, for employees who have a high perception of CSR, the indirect effect of ESSL on EWGB through green role modeling is stronger. Additionally, the effect of ESSL on EWGB through green role modeling is weak when employees' perceptions of CSR are low. Given these findings, we further propose the following moderated mediation hypothesis:

**Hypothesis 5 (H5):** *Through green role modeling, employees' perception of CSR moderates the strength of the mediating relationship between the environmentally specific servant leadership and employee workplace in-role (H5a) and extra-role green behavior (H5b), such that the mediating relationship is stronger when employees perceive CSR.*

## 3. Methods

### 3.1. Sample and Procedures

Our data were collected from the employees of eight companies in Shandong Province, China. The companies we selected for this study have implemented green policies and practices within their organizations, making them appropriate choices for investigating the relationship between ESSL and EWGB. We contacted top management first to explain the purpose for our study and to introduce the data collection procedure. After obtaining

support from the top management of these eight companies, we visited the companies on working days for data collection during the first two weeks of May 2022 for first time (time 1) and the first two weeks of June 2022 for the second time (time 2).

To alleviate potential common method bias, we adopted a two-wave time-lag design for data collection, i.e., survey data were gathered at two different times within one month. Furthermore, to ensure confidentiality and anonymity, the employees were invited and instructed to seal the completed questionnaires in envelopes and to return them directly to the researchers [57]. Specifically, at time 1 (T1), the participants were asked to report on the ESSL of their direct leader, the extent to which they viewed their leaders as green role models, and their own demographic characteristics (see Appendix A). One month later, at time 2, respondents were invited to complete questionnaires about green behaviors at the workplace and perceived CSR of their organizations (see Appendix B).

A total of 512 employees were invited to participate in our study, and at time 2, a total of 450 questionnaires were obtained. Finally, after eliminating invalid data, a sample of 372 valid participants was selected and used in this study, yielding a response rate of 72.7%. For the final sample of 372 respondents, 209 (56.1%) were female and 163 (43.9%) were male. Almost half of the respondents were between 20 and 30 years old, with 48.9% (182) of the survey being in this age group. For work experience, 128 (34.4%) had less than 3 years of work experience, 140 (37.6%) of the respondents had 4–6 years of work experience, and the remaining 104 (28%) had more than 7 years of work experience in their respective organizations. The majority of the respondents had been well educated, with 70.2% having at least a bachelor's degree.

### 3.2. Measures

All questionnaires used in our study were originally written in English and have been validated in previous studies. We translated the English version of the scales to Simplified Chinese using a traditional translation–back translation procedure [58]. A seven-point Likert scale (1 = strongly disagree; 7 = strongly agree) was used for all items in the questionnaire.

### 3.2.1. Environmental-Specific Servant Leadership

We used a 11-item scale adapted from Luu's [11] ESSL scale and the [59] servant leadership scale by Liden et al. to estimate ESSL. A sample item for ESSL was "My supervisor encourages me to contribute to eco-initiatives". Cronbach's alpha of the scale was 0.93.

### 3.2.2. Green Role Modeling

We measured the level of green role modeling using three items adapted from Ogunfowora's [60] well-validated ethical role model perception measure. To assess the extent to which employees viewed their leaders as green role models, we adapted the items by replacing the phrase "ethical" with "green" in each item. A sample item for green role modeling is "My supervisor provides a good green model for me to follow". Cronbach's alpha of the scale was 0.94.

### 3.2.3. Employees' Perceived CSR

Similar to other studies on CSR in China [6,9], we measured employees' perceptions of CSR at the individual level using Turker's [61] scale. In line with Tian and Robertson's [9] research, we removed items about corporate social responsibility toward employees, as some studies pointed out that this dimension should not be considered as part of CSR [62,63]. The scale we finally applied contained 12 items. A sample item is "Our company contributes to campaigns and projects that aim to promote the well-being of the society." The Cronbach's alpha of the scale was 0.90.

3.2.4. Employee Workplace In-Role Green Behaviors and Extra-Role Green Behaviors

Two 3-item scales, adapted from Bissing-Olson et al. [64]—daily task-related pro-environmental behavior at work and daily proactive pro-environmental behavior at work—were utilized to assess in-role EWGB and extra-role EWGB, respectively. A sample item for in-role EWGB is "Today, I fulfilled responsibilities specified in my job description in environmentally friendly ways", and one for extra-role EWGB is "Today, I did more for the environment at work than I was expected to". Cronbach's alphas for in-role and extra-role EWGB were 0.87 and 0.75, respectively.

3.2.5. Control Variables

Some studies have pointed out that demographic variables may influence employee green behaviors [65], so we controlled for employees' gender, age, education level, and organizational tenure in our study.

**4. Data Analysis and Results**

*4.1. Descriptive Analyses*

The means, standard deviations and correlations of the study variables are presented in Table 1. As seen in Table 1, ESSL was positively related to both types of green behavior (in-role, r = 0.53, $p < 0.01$; extra-role, r = 0.64, $p < 0.01$). In addition, ESSL was positively associated with green role modeling (r = 0.78, $p < 0.01$). Furthermore, green role modeling was positively related to both types of green behavior (in-role, r = 0.58, $p < 0.01$; extra-role, r = 0.63, $p < 0.01$). Additionally, CSR was positively related to ESSL (r = 0.74, $p < 0.01$) and green role modeling (r = 0.70, $p < 0.01$).

**Table 1.** Means, standard deviations and correlations of study variables (N = 372).

|  | Mean | SD | 1 | 2 | 3 | 4 | 5 | 6 | 7 | 8 | 9 |
|---|---|---|---|---|---|---|---|---|---|---|---|
| 1. Age | 2.56 | 0.63 | - | | | | | | | | |
| 2. Gender | 1.56 | 0.5 | 0.03 | - | | | | | | | |
| 3. Tenure | 2.01 | 0.93 | 0.60 ** | −0.03 | - | | | | | | |
| 4. Education | 2.72 | 0.76 | 0.04 | 0.10 | 0.09 | - | | | | | |
| 5. ESSL | 5.34 | 0.86 | −0.02 | 0.04 | 0.09 | 0.09 | - | | | | |
| 6. GRM | 5.31 | 1.11 | −0.08 | 0.06 | 0.05 | 0.16 ** | 0.78 ** | - | | | |
| 7. CSR | 5.73 | 0.66 | −0.02 | 0.03 | 0.11 * | 0.15 ** | 0.74 ** | 0.70 ** | - | | |
| 8. ERGB | 5.51 | 0.77 | 0.04 | 0.06 | 0.16 ** | 0.21 ** | 0.64 ** | 0.63 ** | 0.61 ** | - | |
| 9. IRGB | 5.35 | 0.83 | 0.02 | 0.01 | 0.12 * | 0.18 ** | 0.53 ** | 0.58 ** | 0.55 ** | 0.64 ** | - |

Note: ESSL = environmentally specific servant leadership; GRM = green role modeling; CSR = corporate social responsibility; IRGB = in-role green Behavior; ERGB = extra-role green behavior; * $p < 0.05$; ** $p < 0.01$.

*4.2. Confirmatory Factor Analysis*

We performed a confirmatory factor analysis of structural equation modeling using the Amos graphical approach. To be specific, we first examined the five-factor model, including five latent variables: ESSL, GRM, CSE, ERGB and IRGB. The fit index in Table 2 shows that the five-factor model fit the data well ($\chi^2(265) = 555.94$; CFI = 0.96; TLI = 0.95; IFI = 0.96; RMSEA = 0.05). Second, we compared the fit indices for different models. As we can see from Table 2, the results of the model comparison revealed that the five-factor model fit the data significantly better than alternative models, including a four-factor model combining in-role green behavior and extra-role green behavior into a single factor ($\chi^2(269) = 659.98$; CFI = 0.95; TLI = 0.94; IFI = 0.95; RMSEA = 0.06), a three-factor model in which variables were merged based on their measurement order over time ($\chi^2(272) = 1016.43$; CFI = 0.90; TLI = 0.89; IFI = 0.90; RMSEA = 0.08) and a single-factor model with all variables combined into one ($\chi^2(275) = 1494.93$; CFI = 0.83; TLI = 0.82; IFI = 0.83; RMSEA = 0.10). These findings support the idea that these five factors are different constructs in this study.

**Table 2.** Model fit index.

| Model | X2 | DF | CFI | TLI | IFI | RMSEA |
|---|---|---|---|---|---|---|
| 5 factor | 555.94 | 265.00 | 0.96 | 0.95 | 0.96 | 0.05 |
| 4 factor | 659.98 | 269.00 | 0.95 | 0.94 | 0.95 | 0.06 |
| 3 factor | 1016.43 | 272.00 | 0.90 | 0.89 | 0.90 | 0.08 |
| 1 factor | 1494.93 | 275.00 | 0.83 | 0.82 | 0.83 | 0.10 |

*4.3. Hypotheses Testing*

Hypothesis 1a,b suggest that ESSL has a positive effect on both IRGB and ERGB. To compare whether these two strengths are the same, we performed structural equation modeling using the Amos graphical method. As shown in Figure 2, our model fits the data well ($\chi^2$ (165) = 452.14; CFI = 0.95; TLI = 0.95; IFI = 0.94; RMSEA = 0.068). Additionally, we can find that ESSL is positively related to both IRGB ($\beta$ = 0.19, $p < 0.05$) and ERGB ($\beta$ = 0.45, $p < 0.001$). Furthermore, ESSL is positively related to GRM ($\beta$ = 0.83, $p < 0.001$). These results support Hypotheses 1a,b and 2.

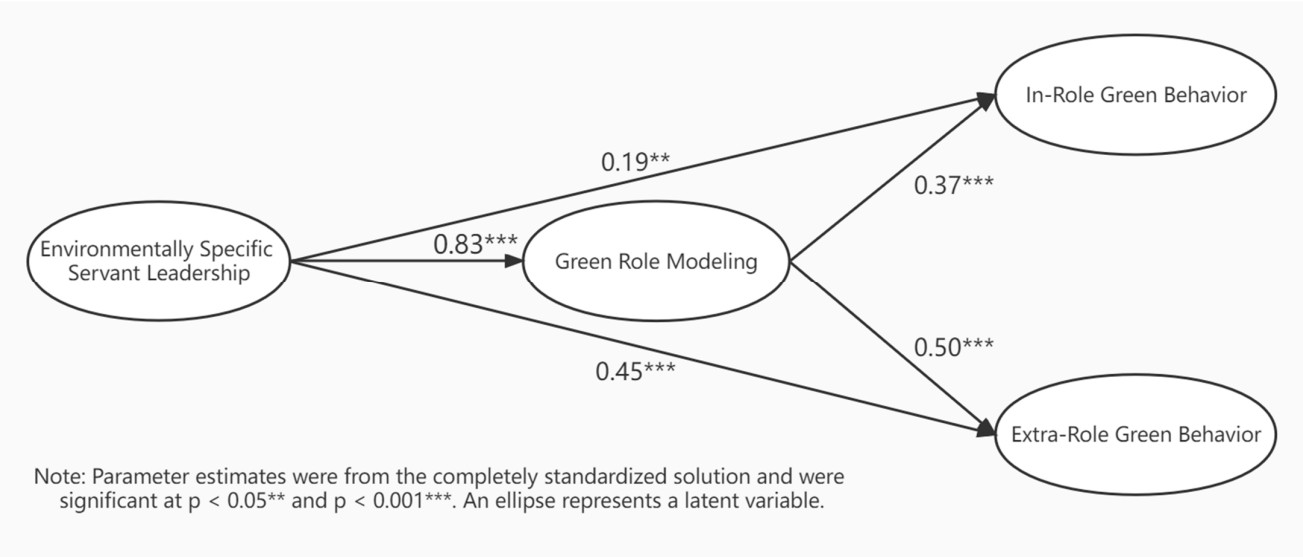

**Figure 2.** Proposed theoretical model with standardized path coefficients.

*4.4. Mediation Analyses*

The bootstrapping approach suggested by Preacher and Hayes [66] was conducted to test the significance of the mediating effect of GRM on the relationship between ESSL and EWGB. To be specific, if the 95% bias-corrected confidence interval shown by the bias-corrected bootstrap procedure is not zero, then we consider the indirect effect to be significant. After creating 5000 bootstraps, we found a significant mediating effect of GRM on the relationship between ESSL and employee workplace in-role green behaviors ($\beta$ = 0.32). The 95% BC confidence intervals for the indirect effects were between 0.21 and 0.43, which did not overlap with zero ($p < 0.05$). Similarly, the mediating effect of GRM on the relationship between ESSL and employee workplace extra-role green behaviors is significant ($\beta$ = 0.22). The 95% BC confidence intervals for the indirect effects were between 0.13 and 0.32 ($p < 0.05$). Therefore, Hypothesis 3a,b were supported.

*4.5. Moderation and Moderated Mediation Analysis*

First, we used the PROCESS macros Model 1 by Hayes [67] to test the moderating effect of CSR on the relationship between ESSL and EWGB. As shown in Table 3, the interaction between GRM and CSR was significant (IRGB: B[SE] = 0.07[0.03], $p < 0.05$; ERGB: B[SE] = 0.14[0.04], $p < 0.01$).

**Table 3.** Results of the moderating effects of CSR.

| | IRGB | | | ERGB | | |
|---|---|---|---|---|---|---|
| **Predictor** | **B** | **SE** | **Predictor** | **B** | **SE** | |
| GRM | −0.10 | 0.18 | GRM | −0.51 | 0.21 | |
| CSR | 0.04 | 0.16 | CSR | −0.34 | 0.19 | |
| GRM * CSR | 0.07 * | 0.03 | GRM * CSR | 0.14 ** | 0.04 | |
| $R^2$ | 0.70 | | | 0.64 | | |

Note: N = 372; * $p < 0.05$; ** $p < 0.01$; GRM = green role modeling; CSR = corporate social responsibility; IRGB = in-role green behavior; ERGB = extra-role green behavior.

Furthermore, Figure 3 shows the moderating effect of CSR at two levels of standard deviation ± 1. As we can see, the relationship between ESSL and EWGB was stronger for employees who perceived their leaders to be green role models. Therefore, Hypothesis 4a,b were supported.

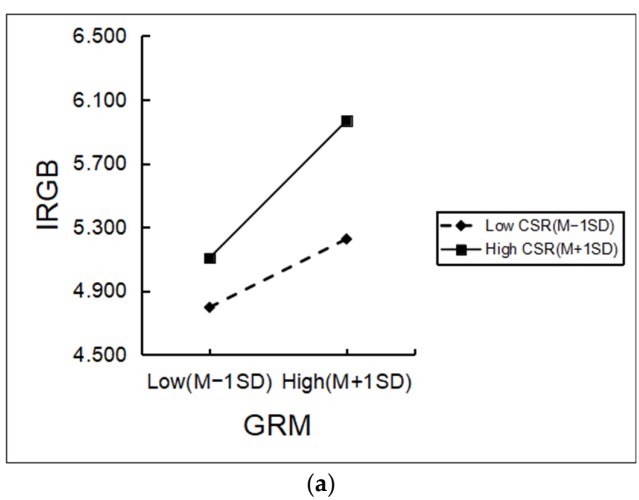

(**a**)

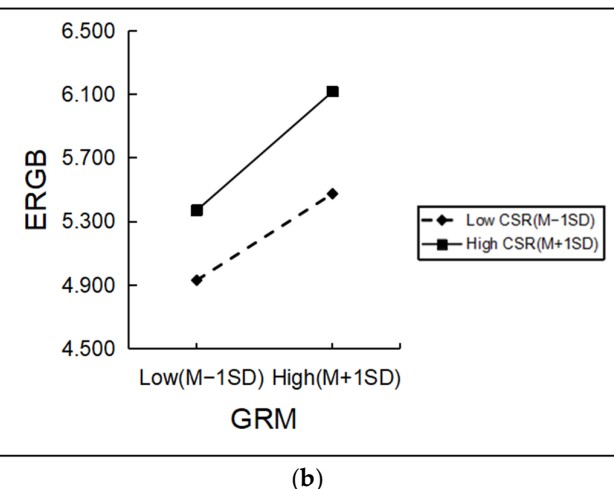

(**b**)

**Figure 3.** The moderating effect of CSR. (**a**) IRGB = in-role green behavior; (**b**) ERGB = extra-role green behavior; GRM = green role modeling; CSR = corporate social responsibility.

The PROCESS macro allows for tests on moderated mediating effects, so we followed the method of Preacher et al. using PROCESS macro Model 14 by Hayes [67] to test the moderating mediating effects. As shown in Table 4, the moderating effect of CSR on the relationship between ESSL and EWGB was significant. Overall, the results of all hypotheses testing are presented in Table 5.

Our results for all hypotheses tests are shown in Table 5.

**Table 4.** Results of moderated mediation analysis.

| | Indirect Effect (β, Boot SE) | 95% CI (Lower-Level CI, Upper-Level CI) |
|---|---|---|
| | IRGB | |
| −1 SD | 0.19(0.07) | (0.0493,0.3428) |
| M | 0.29(0.06) | (0.1682,0.4182) |
| +1 SD | 0.38(0.08) | (0.2409,0.5407) |
| Index of moderated mediation | 0.15(0.06) | (0.0244,0.2605) |
| | ERGB | |
| −1 SD | 0.14(0.06) | (0.0348,0.2534) |
| M | 0.19(0.05) | (0.0968,0.2906) |
| +1 SD | 0.25(0.06) | (0.1226,0.3608) |
| Index of moderated mediation | 0.08(0.05) | (0.0236,0.1640) |

Note: N = 372, the reported regression coefficients are not standardized and were computed with 2000 bootstrap samples. IRGB = in-role green behavior; ERGB = extra-role green behavior.

**Table 5.** Results of hypotheses.

| Hypothesis | | Result |
|---|---|---|
| 1a | ESSL is positively related to employee workplace in-role green behavior. | Supported |
| 1b | ESSL is positively related to employee workplace extra-role green behavior. | Supported |
| 2 | ESSL is positively related to green role modeling. | Supported |
| 3a | Green role modeling mediates the relationship between ESSL and employee workplace in-role green behavior. | Supported |
| 3b | Green role modeling mediates the relationship between ESSL and employee workplace extra-role green behavior. | Supported |
| 4a | Employees' perception of CSR moderates the relation between green role modeling and employee workplace in-role green behavior. | Supported |
| 4b | Employees' perception of CSR moderates the relation between green role modeling and employee workplace extra-role green behavior. | Supported |
| 5a | Employees' perception of CSR moderates the strength of the mediating relationship between the ESSL and employee workplace in-role green behavior through green role modeling. | Supported |
| 5b | Employees' perception of CSR moderates the strength of the mediating relationship between the ESSL and employee workplace extra-role green behavior through green role modeling. | Supported |

## 5. Discussion and Implications

Faced with growing environmental problems, organizations have started to implement green policies and to take actions for environmental protection and sustainable development [68]. EWGB as an important factor in corporate sustainable development initiatives has received increasing attention from both scholars and practitioners [9,11,12]. In this context, our research aims to explore how and when employees would emulate their leaders' green behaviors. Based on the results of the empirical study, we found that employees imitate the pro-environmental behaviors of their leaders via role modeling; meanwhile, CSR plays a moderating role in this process. Our findings are specified as follows:

To begin, the finding that ESSL has a positive effect on both employee in-role and extra-role workplace green behaviors is consistent with that of previous literature [6,12,17]. Moreover, our finding suggests that ESSL is indirectly related to both in-role and extra-role green behaviors through the mediation of employees' perceived green role models. In other words, according to social learning theory, we suggest that the more leaders demonstrate ESSL, the more employees are likely to view them as green role models and to exert more in-role and extra-role green behaviors. Next, the finding suggests that employees' perceived CSR moderates the mediating relationship between ESSL and EWGB through employees' perceived green role models. Specifically, when employees perceive a high degree of perceived CSR, the mediated relationship between ESSL and EWGB through the perception of leaders as green role models was stronger. We believe that managers' environmentally specific servant behaviors and organizational CSR policies and practices could promote employees' workplace green behaviors, which contributes to the sustainability, environmental protection and societal development of society as a whole. The theoretical and practical implications of our findings are discussed below.

### 5.1. Theoretical Implications

Through these findings, our study makes theoretical contributions to the growing body of research on green leadership and management in the following four ways.

First, as a response to Robertson and Barling's [13] call for research on environmentally specific servant leadership, the findings of our study expanded understanding of ESSL and confirmed the positive impact of ESSL on EWGB. A large number of studies in the existing literature focuses on the influence of leadership style on employee green behavior [69,70],

but not enough attention has been paid to ESSL, a comparatively newly defined leadership style [17]. Our study confirms that endeavors by an ESS leader who is willing to support and serve followers in green ways can increase employees' green behaviors. Moreover, previous studies mainly examined the positive impact of environmentally specific servant leadership on employee extra-role green behaviors [9,71] rather than on employee in-role green behaviors. However, our study fully examined the positive impact of ESSL on both employee in-role and extra-role green behaviors to obtain a complete and comprehensive understanding of employee green behaviors. Servant leaders who are humble always make the development of others one of their goals at work [28]. Thus, employees will have a much easier time accepting the work instructions from ESS leaders and behaving in a green way in the workplace. Our findings confirmed that ESSL not only positively influences an employee's extra-role green behaviors, which echoes the findings of several studies in the literature, but also facilitates the employee's in-role green behaviors as well.

Second, this study extends the current literature by demonstrating the mediating effect of employees' perception of their leaders' green role modeling on the relationship between ESSL and EWGB. Previous literature generally suggests that employees learn and imitate the green behaviors of their leaders through observational learning [23]. Although a large number of studies have noticed this mechanism [17,71], empirical testing of this process is still lacking. In terms of a social learning perspective, our findings indicate that ESSL increases employees' perceptions of their leaders as green role models and, consequently, increases their own in-role and extra-role green behaviors. Furthermore, the adoption of social learning theory will definitely enrich the theoretical boundaries of social learning theory and provide a comprehensive mechanism through which ESSL influences EWGB.

Third, drawing on social learning theory [21], our research also contributes to EWGB research by exploring the boundary conditions in the indirect effect of ESSL on EWGB. As CSR is regarded as an effective strategy for achieving sustainability, a number of scholars have examined the role of CSR in their studies of employee green behavior [9,72]. Our study is the first to examine the moderating effect of employees' perceived CSR in the ESSL literature. We provide a theoretical model and utilize empirical evidence to explain why employees' CSR perceptions moderate the mediating relationship between ESSL and EWGB through green role modeling. We found that, for employees' who perceive high levels of CSR, high scores on ESSL lead to a greater green role modeling effect, thus increasing employees' green behaviors.

Fourth, the current study enhanced our understanding of ESSL and EWGB in the Chinese context. By drawing from theories regarding ESSL and EWGB developed in the Western context, we found that the results are consistent with what is expected in a Western context, which emphasizes the universal aspects of the relevant concepts and theories.

*5.2. Practical Implications and Policy Suggestions*

Green management is a contemporary global concern, and our study on ESSL in encouraging and motivating EWGB has important managerial implications. The findings of our model suggest that if enterprises wish to inspire green behaviors of their subordinates, an effective way is to actively take on corporate social responsibility and encourage leaders to act in a green manner so that they can serve as green role models by their subordinates.

More specifically, organizations should estimate, develop and reward ESS leadership behaviors. For instance, more attention should be paid to the recruitment and promotion process during which the organizations select potential leaders who exhibit ESS leadership. Moreover, the traits of ESS leadership could be added to the criteria for a performance appraisal system and a reward system for these leaders. For example, the organization could give extra bonuses to leaders who promote electricity conservation among their subordinates. In addition, we suggest that the HRM department consider providing relevant training programs to leaders for ESS leadership development. This suggestion regarding HRM practices can help leaders emphasize ESS leadership behaviors and act as green role models.

Furthermore, our research also points out that employees' green behaviors are influenced by their perceptions of their leaders. Therefore, organizations should pay equal attention to leaders and subordinates during the implementation of green policies. For example, in training process, organizations should encourage subordinates to discover the good qualities of their leaders and regard them as green role models, which can help to facilitate the social learning process of their employees.

Finally, as the results of model testing indicated, a high level of perceived CSR can promote employee workplace in-role and extra-role green behaviors because the employees can learn directly from organizational CSR values and actions. Therefore, we believe that organizational CSR practices can help create a pro-social and pro-environmental atmosphere and climate within the organization. These CSR practices, which are beneficial for individual and team development, trigger leaders to exhibit servant behaviors toward environmental goals. Thus, it is essential for organizations to adopt CSR practices to create a "green" climate to motivate not only the leaders but also the employees. In summary, organizations should adopt specific HRM practices and CSR behaviors to increase the "green" climate to increase employees' green behaviors.

Based on the results of this study, the following policy recommendations are proposed: The government should encourage enterprises to adopt CSR behaviors, implement relevant management practices and engage in environmentally friendly activities. On the one hand, we suggest that the Chinese government should target distinct policy arrangements toward enterprises with different types of ownership. First, it is important to develop a supervision mechanism to monitor the CSR-related or environmental behaviors of state-owned enterprises, e.g., whether the enterprise publishes proper annual CSR reports or whether the enterprise has an environmental policy and behaves in an environmentally friendly way. Second, the government could motivate non-state-owned enterprises to play an important role in CSR implementation and environmental protection. On the other hand, the central government should establish a proper performance evaluation mechanism with both financial and non-financial performance standards, e.g., environmental standards for local governments. Meanwhile, the government could appeal to relevant non-governmental or non-profit organizations to participate in monitoring enterprises' green behaviors.

*5.3. Limitations and Future Research Directions*

Despite the fact that this study makes theoretical contributions to the literature and practical implications to practitioners, it still has some limitations that should be noted. First, all the variables in our study were self-reported by employees in the data collection process, which perhaps leads to common method bias. Although common method bias was avoided to some extent in our study due to the two-wave time-lag design [57], common method bias cannot be completely eliminated. Therefore, we suggest that future studies could select data from multiple sources for their studies, such as supervisors being invited to rate employees' green behaviors. Moreover, the adoption of employee self-reports for data collection may suffer from a lack of objectivity. Although a meta-analysis pointed out that self-reported data are correlated with actual data in terms of green behavior [73], we recommend scholars utilize more objective data for future studies to validate the study. Second, this study considers employees' perceived green role model of leadership as mediating factors. However, from the perspective of social learning theory [21], we believe that other factors also affect the learning effectiveness of employees. Thus, we suggest that future studies consider other factors at the organizational, team and individual levels as mediating factors for the study. Third, our study only considered CSR as a boundary condition that influences the social learning process of employees' green behaviors. We believe that more variables have an impact on the social learning process of employees, and future studies can consider variables such as organizational identity, individual green values, and green self-efficacy. Fourth, we have only tested our theory in the Chinese context, and future studies may consider extending our theory to different cultural contexts for validation.

## 6. Conclusions

The purpose of this study is to examine when employees emulate the green behaviors of their leaders and to empirically test how environmentally specific servant leadership (ESSL) affects employee workplace green behaviors (EWGB) from a social learning perspective. The results showed the mediating effects of green role modeling in the relationship between ESSL and the employees' in-role and extra-role green behaviors. Moreover, the findings also indicated the moderating effect of employees' perceived CSR on the mediating relationship between environmentally specific servant leadership, and employee workplace in-role green behaviors and employee workplace extra-role green behaviors through green role modeling. Such findings filled the research gap by adopting social learning theory to examine the mechanism of how ESSL affects EWGB. Our results demonstrate that in order to successfully implement an environmental sustainability strategy, organizations must develop a new generation of "green" leaders, and to do so, we suggest that the organizations could incorporate effective green policies in their human resource management frameworks in order to achieve better green performance. While this study only focused on the social learning process at an individual level, we hope that future research extends this study to the group level.

**Author Contributions:** Conceptualization, F.G.; Formal analysis, F.G.; Funding acquisition, F.G.; Investigation, F.G.; Methodology, F.G.; Project administration, F.G.; Software, J.L.; Validation, J.L.; Writing—original draft, J.L.; Writing—review & editing, F.G. All authors have read and agreed to the published version of the manuscript.

**Funding:** This study was supported by the Fundamental Research Funds for the Central Universities (CUPL: 20ZFQ63001), the Social Science Foundation from the China University of Political Science and Law (10820366 and 20ZFQ63001), the National Science Foundation of China (71874205) and the National Social Science Foundation of China (20AZD071).

**Institutional Review Board Statement:** Not applicable.

**Informed Consent Statement:** Not applicable.

**Data Availability Statement:** The data collected to support the findings of the study are available from the corresponding author upon request.

**Conflicts of Interest:** The authors declare no conflict of interest with respect to the research, authorship and/or publication of this article.

## Appendix A. Questionnaire Used in This Study (Time 1)

Dear Sir/Madam.

Hello! We are the researchers of Business School of China University of Political Science and Law, thank you very much for helping us to complete this questionnaire in your busy schedule.

The purpose of this survey is to understand some of your feelings, thoughts and personal characteristics at work, the results of the questionnaire are for academic research purposes only, we will ensure that the information you fill in will not be disclosed to others.

The following are some scenarios or questions related to your work, the answers are not good or bad, please give your feedback on the corresponding items according to your real situation. We respectfully invite you to answer each question in the questionnaire, and to choose only one answer for each question. The conduct of this study relies on your assistance and support, and we would like to express our sincere gratitude!

Basic information (Please circle the appropriate answer according to your actual situation)
Gender: Male Female
Education: Below college; College Bachelor's degree; Master's degree; Doctoral degree
Age: below 20; 20–30; 31–40; 41–50; 51 and above
Organizational Tenure: 3 years and below; 4–6 years; 7–10 years; 11 years and above.

Please circle the one number for each question that comes closest to reflecting your perception about it where 1 means strongly disagree and 7 means strongly agree

Section 1: Measures of Environmentally Specific Servant Leadership [11,59]

1. My supervisor cares about my eco-initiatives.
2. My supervisor emphasizes the importance of contributing to the environmental improvement.
3. My supervisor is involved in environmental activities.
4. I am encouraged by my supervisor to volunteer in environmental activities.
5. My supervisor has a thorough understanding of our company and its environmental goals.
6. My supervisor encourages me to contribute eco-initiatives.
7. My supervisor gives me the freedom to handle environmental problems in the way that I feel is best.
8. My supervisor does what she/he can do to realize my eco-initiatives.
9. My supervisor holds high environmental standards.
10. My supervisor always displays green behaviors.
11. My supervisor would not compromise environmental principles in order to achieve success.

Section 2: Measures of Green Role Modeling [60]

12. My supervisor provides a good green model for me to follow.
13. My supervisor leads by example on green behaviors.
14. My supervisor providing an appropriate green model.

Note: Question 1 to question 11 were adapted from Luu's [11] ESSL scale and Liden et al.'s [59] servant leadership scale. Question 12 to question 14 were adapted from Ogunfowora's [60] ethical role model perception scale.

## Appendix B. Questionnaire Used in This Study (Time 2)

Dear Sir/Madam.

Hello! We are the researchers of Business School of China University of Political Science and Law, thank you very much for helping us to complete this questionnaire in your busy schedule.

The purpose of this survey is to understand some of your feelings, thoughts and personal characteristics at work, the results of the questionnaire are for academic research purposes only, we will ensure that the information you fill in will not be disclosed to others.

The following are some scenarios or questions related to your work, the answers are not good or bad, please give your feedback on the corresponding items according to your real situation. We respectfully invite you to answer each question in the questionnaire, and to choose only one answer for each question. The conduct of this study relies on your assistance and support, and we would like to express our sincere gratitude!

Section 1: Measures of Green Behavior [64]

In role

1. Today, I adequately completed assigned duties in environmentally friendly ways.
2. Today, I fulfilled responsibilities specified in my job description in environmentally friendly ways.
3. Today, I performed tasks that are expected of me in environmentally friendly ways.

Extra role

4. Today, I took a chance to get actively involved in environmental protection at work.
5. Today, I took initiative to act in environmentally friendly ways at work.
6. Today, I did more for the environment at work than I was expected to.

Section 2: Measures of Perceived CSR [61]

7. Our company participates in activities, which aim to protect and improve the quality of the natural environment.
8. Our company makes investment to create a better life for future generations.

9. Our company implements special programs to minimize its negative impact on the natural environment.

10. Our company targets sustainable growth, which considers future generations.

11. Our company supports nongovernmental organizations working in problematic areas.

12. Our company contributes to campaigns and projects that aim to promote the well-being of the society.

13. Our company encourages its employees to participate in voluntary activities.

14. Our company respects consumer rights beyond the legal requirements.

15. Our company provides full and accurate information about its product to its customers.

16. Customer satisfaction is highly important for our company.

17. Our company always pays its taxes on a regular and continuing basis.

18. Our company complies with legal regulations completely and promptly.

Note: Question 1 to question 6 were adapted from Bissing-Olson et al.'s [64] daily task-related pro-environmental behavior at work and daily proactive pro-environmental behavior at work scale. Question 7 to question 18 were adapted from Turker's [61] employees' perceptions of CSR at the individual level scale.

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
