# Peer review of "Environmentally Specific Servant Leadership and Employee Workplace Green Behavior: Moderated Mediation Model of Green Role Modeling and Employees’ Perceived CSR"

_sustainability, doi:10.3390/su141911965_

Round 1

Reviewer 1 Report

Reviewer #2: Reviewer comment (R.C.): The abstract needs to be improved. Your Abstract should answer these questions about your manuscript: What was done? Why did you do it? What did you find? Why are these findings useful and important? Answering these questions lets readers know the most important points about your study and helps them decide whether they want to read the rest of the paper. Make sure you follow the proper journal manuscript formatting guidelines when preparing your abstract.
Reply: 
Reviewer comment (R.C.): The keywords list needs to be improved. Reply:

Reviewer comment (R.C.): The introduction needs to be improved, and it is necessary to expose the gap in the literature regarding the topic of investigation, the motivation, relevance,  innovation, and contribution,  that this study brings compared with others that existed. 

Reply:
Reviewer comment (R.C.):  The authors need to add the following investigations to the manuscript's body: 
Analyzing the environmental Kuznets curve for the EU countries: the role of ecological footprint. Environmental Science and Pollution Research volume 25, pages29387-29396 (2018). https://doi.org/10.1007/s11356-018-2911-4 
Causal relationship between CO2 emissions, real GDP, energy consumption, financial development, trade openness, and urbanization in Tunisia. Environmental Science and Pollution Research volume 22, pages15663-15676 (2015). https://doi.org/10.1007/s11356-015-4767-1. 
article/10.1007/s10669-022-09846-2. 
economies10060131. 

These articles will complement the literature review and improve the manuscript's quality. 
Reviewer comment (R. C.):  I recommend that the authors need to give a justification regarding the choice of the object of study, variables, and time series. These justifications will facilitate the reapplication of this study by other authors. 

Reviewer 2 Report

English must be edited completely before review.

Reviewer 3 Report

Dear authors,

I appreciate having the opportunity to review the manuscript entitled “Environmentally-specific Servant Leadership and Employee Workplace Green Behavior: Moderated Mediation Model of Green Role Modeling and Employees’ Perceived CSR” (Manuscript ID: sustainability-1841936).

Based on a social learning perspective, this research investigated the mediating effect of green role modeling and the moderating effect of employees’ perceived CSR on the relationship between ESSL and EWGB. I belive that the authors have made considerable efforts to develop this paper, thus, the current version of manuscript is required to be revised in a minor manner. I want to provide some suggestions for the improvement of this paper as follows.

[1] Introduction

- I think that the overall structure and writing of introduction part are clear and well-aligned because it is easy to catch what the research questions or and strategies to deal with are in this paper. However, would you please add one or two more research gaps and its resolving processes in the introduction part?

[2] Theories and hypotheses

- Based on the interesting phenomena that this paper caught, it provides adequate theoretical background and support for the development of its hypotheses. 

[3] Method

- The authors described that this research conducted a structural equation modeling (SEM) and PROCESS macro analysis. However, I am not sure whether the authors utilized the SEM or simple path analysis. In Figure 1 in your manuscript, all variables are presented as a form of rectangular. As the authors already knew, the rectangular form is used to express the results of path analysis, not SEM. Please clearly describe this issue. 

 I wish these comment may help you to improve your paper. Good luck.

Reviewer 4 Report

I have the following observations, questions, and comments that may help to improve your work. The authors must modify the following points in great detail. 
1. In the abstract, please include 2-3 special quantitative achievements from the findings of this study in the context of the environment by combining the research objectives and problems. Check spellings for many words that are misspelt or written in haste. 
2. The introduction section needs a few more sentences to strengthen the article, and please include the research problem, objective, and novelty in the last paragraph of the Introduction section. 
3. Include a few more sentences at the beginning of the introduction explaining your paper's contribution to the environment, climate change impact, and sustainability, as well as your attempts to deal with or present solutions to a specific problem/s and your unique contribution with this research paper. 
4. Please also present the methodology section in a concise graphical format. 
5. The literature review section is very weak; please revise it. 
6. Please present your literature review in the form of a SmartArt chart. 
7. Just after the Methodology, please mention the societal benefits of your research in terms of evaluating its key determinant. 
8. In 500-750 words, explain research problems, solutions, and the theoretical contribution of your study in the "Results" section. 
9. Please include graphical presentations of your findings. 
10. Describe why you placed this study in a separate section of "Policy Suggestions" just before the section of "Conclusions." 
I found that the literature section is a little weak, shift your study a little more towards environment friendly and sustainability, therefore it requires more studies to be reviewed therefore I suggest you to include the following work:
https://doi.org/10.1016/j.jhtm.2020.11.008

https://doi.org/10.1007/s10668-020-01163-5

https://doi.org/10.1108/JGR-12-2018-0088

https://doi.org/10.1016/j.ijpe.2021.108393

 Looking forward for your revised submission. 

Round 2

Reviewer 1 Report

no further comments.

Reviewer 2 Report

the paper in present form is a normal and moderate paper for publishing in this journal. The just point is that if the decision of Editor in chief is publishing this paper quality of all figures must be improved. 

Reviewer 4 Report

Dear Authors,

The paper is improved but sincerely the conclusions are not clear. I suggest to enhance the entire-section providing managerial implications, main issues and what are the key-messages of your work.
